# Overcoming the Tropical Andes publication divide: Insights from local researchers on challenges and solutions

Jose W. Valdez [1,2]☯*, Lucía Castro Vergara [3]☯*, Gabriela Orihuela[3], Miguel Fernandez[1,2,4,5]

**1** German Centre for Integrative Biodiversity Research (iDiv) Halle-Jena-Leipzig, Leipzig, Germany, **2** Institute of Biology, Martin Luther University Halle Wittenberg, Halle (Saale), Germany, **3** Asociación para la Conservación de la Cuenca Amazónica - ACCA, Miraflores, Lima, Perú, **4** Department of Environmental Science and Policy, George Mason University, Fairfax, VA, United States of Ameirca, **5** Instituto de Ecología, Universidad Mayor de San Andrés, La Paz, Bolivia

☯ These authors contributed equally to this work.
* jose.valdez@idiv.de (JWV); lcastro@conservacionamazonica.org (LCV)

**Data Availability Statement:** All relevant data are within the manuscript and its Supporting Information files.

## Abstract

The Tropical Andes, one of the world's most biodiverse regions, is vital for ecological research and conservation. However, while researchers in Bolivia, Ecuador, and Peru contribute significantly to scientific knowledge, their publication rates in academic journals have historically lagged behind neighboring nations. A multifaceted strategy was employed to understand and address the publication divide in the Tropical Andes region. This approach focused on regional researchers and consisted of a three-day workshop to improve scientific writing skills, offer publication insights, and equip researchers with tools to overcome obstacles. A series of surveys were also conducted to explore the challenges faced by local researchers and their proposed solutions, covering topics such as participant demographics, factors contributing to lower publication rates, personal barriers, proposed strategies for improving publications, specific topics of interest, participant satisfaction, most valuable workshop topics, and future recommendations. The workshop had an overwhelming response, with over 500 interested participants registering in just a few days, mostly experienced professionals, highlighting the need for such initiatives in the region. About two-thirds had ready-to-publish materials, highlighting the potential impact of targeted interventions on unlocking untapped knowledge. The surveys revealed the challenges contributing to the publication divide, including insufficient training, cultural emphasis on economic development, language barriers, limited resource access, lack of institutional support, high publishing costs, and time and financial constraints. The most common personal barriers were insufficient knowledge and experience in the publication process, lack of self-confidence, and fears of rejection. Proposed solutions include conducting training workshops, fostering collaborative networks, improving resource accessibility, and an institutional and cultural shift that encourages publishing. Addressing challenges faced by experienced professionals in the Tropical Andes by understanding individual needs, fostering support, and demystifying the publication process offers a

**Funding:** This research received funding from the ERANet Joint Call 2016–2017 (DLR Förderkennzeichen 01DN19032 Tropical Andes Observatory—TAO). It was also supported by the German Centre for Integrative Biodiversity Research (iDiv) Halle-Jena-Leipzig, funded by the German Research Foundation (FZT 118, 202548816). The funders had no role in study design, data collection and analysis, decision to publish, or preparation of the manuscript.

**Competing interests:** The authors have declared that no competing interests exist.

promising path to closing the publication divide and unlocking the region's valuable scientific contributions.

## Introduction

The Tropical Andes region, spanning several countries in South America, is one of the most biodiverse areas on Earth, with its unique ecosystems harboring the highest number of endemic plant and animal species in the world [1–3]. This exceptional diversity not only holds intrinsic value but also provides vital ecosystem services crucial for local populations and the global community [4, 5]. However, over a quarter of the region's original habitat has been lost to human activities, with ongoing threats from agricultural expansion, mining, infrastructure development and urbanization further disrupting ecosystems and accelerating biodiversity loss [3, 6–8]. Addressing these challenges requires sharing research findings to inform evidence-based policies, targeted interventions, and sustainability efforts. While the scientific landscape of Latin America has been characterized by an explosive growth in research output [9–11], the broader Tropical Andes have not adequately reflected its remarkable biodiversity and conservation importance. Countries like Brazil, Mexico, Argentina, and Chile have emerged as leaders, driving this surge with an exponential increase in the number of scientific publications, regional and Spanish journals, and citations [9, 11–16]. Within the Tropical Andes, Colombia stands out as a prominent exception, emerging as a leader in scientific research and publishing both within Latin America and globally [15, 17–19]. Colombia has become one of the world's pioneers in open-access publishing, with over 60% of its research freely available - the 2nd highest globally [18]. Its dedication to open access data and a strong research culture has elevated the reach, influence, and impact of its scientific output, evident through the proliferation of its work in highly visible journals and rapid growth in its publishing landscape [10, 19]. However, this progress contrasts starkly with the broader Tropical Andes region, highlighting the need for targeted initiatives to bridge this gap and unlock the full scientific potential of this biodiverse area.

Despite geographic proximity and shared biodiversity, nations like Bolivia, Ecuador, and Peru face significant challenges matching Colombia's publication volume and growth rates [17, 20]. Economic factors partially shape this representation gap, with higher GDP and research spending correlating with increased scientific output [9, 14, 16, 17, 19]. Complexities arising from the prioritization of economic development, limited research funding, limited access to resources, cultural barriers, linguistic diversity, and historical disparities, have also collectively hindered the scientific productivity of these countries [5, 7, 21]. This imbalance becomes particularly evident when analyzing publication rates, citation metrics, and the large availability of Spanish and regional journals, revealing a noticeable publication gap between these countries and the broader Latin American region [17, 20, 22, 23]. Previous research based on stakeholder consultations in Bolivia, Ecuador, and Peru has highlighted the publication gap as a major issue with training workshops to build and improve skills in scientific writing and publishing proposed as potential solutions [21]. By identifying the factors driving the publication divide and implementing targeted support, we can empower local researchers to elevate their scientific contributions, amplify their voices within global discourse, and foster collaborations across borders.

In this study, we aim to investigate the underlying factors contributing to the lower research publication output observed in Bolivia, Ecuador, and Peru compared to neighboring countries.

We conducted a comprehensive workshop along with surveys to explore the specific challenges faced by researchers in this region and gain insights into their personal experiences with the scientific publishing process. The primary aim was to gather in-depth perspectives from those actively seeking support and guidance to overcome challenges in publishing their research, identify common themes and limitations, formulate targeted strategies to address these challenges, empower researchers, and create an environment conducive to increased scientific contributions. The workshop was specifically tailored to improve scientific writing skills, offer insights into the publication process, and equip researchers with the necessary tools to overcome obstacles in publishing their work. We also aimed to gather comprehensive information about scientific research in the Tropical Andes, identify limitations and obstacles, evaluate the effectiveness of the workshop intervention, and solicit feedback for future workshops. With this workshop and survey-based study, we aimed to bridge the gap between the biodiversity of the Tropical Andes and the challenges faced by its scientific community, ultimately working towards a more equitable and diverse global scientific landscape.

## Materials and methods

An open call for participation in the workshop was disseminated by Asociación para la Conservación de la Cuenca Amazónica - ACCA (ACCA) on July 14, 2022, via their social media platforms, Facebook and Twitter (S1 Appendix). The call included a link to a registration form for interested participants. Despite a modest social media following and one post on each of the platforms, the call resulted in an overwhelming response with 520 applicants expressing interest in the workshop in just five days. Due to the unexpected high demand and capacity limitations of the virtual platform (Zoom), the registration period was closed after five days to ensure a high-quality workshop experience for a manageable number of participants.

From the pool of over 500 registered applicants, we carefully selected 292 participants to take part in the workshop. The selection process focused on geographical origin (prioritizing the Tropical Andes region, particularly Peru, Bolivia, Ecuador, Colombia, and Venezuela), previous participation in related workshops and those on the ACCA mailing list (indicating relevant interest and engagement), age and academic background (excluding applicants 22 years old or younger, recent undergraduates within the last year and prioritizing those with postgraduate education), academic expertise (favoring ecology and conservation over communication and social sciences), English proficiency, and individuals with material ready for publishing. After thoroughly evaluating survey responses against these criteria, we were able to have a select group of participants with diverse expertise and experiences, enabling us to tailor the workshop to meet their needs and expectations.

### Workshop

The virtual workshop was held in Lima, Peru, over three days from July 21–23, 2022. The primary objectives were to enhance participants' abilities to convert research findings into publishable papers, improve organizational skills, and write effectively in English. Additionally, the workshop aimed to familiarize participants with the publication process, including understanding journal selection and responding to reviewer comments. Each day focused on a specific theme:

- **Day 1**: An introductory session that provided an overview of the workshop's objectives and expectations.

- **Day 2**: "*Why publish and how to overcome obstacles*" focused on understanding the importance of publishing scientific manuscripts and addressing specific obstacles faced by

researchers in the Tropical Andes. Participants identified common barriers and discussed strategies to overcome them.

- **Day 3**: "*Creating a scientific manuscript*", participants delved into the publication process. They received guidance on pre-submission aspects such as cover letters, co-authorship, formatting, and selecting the appropriate journal. The post-submission phase was also covered, including understanding editorial decisions, responding to reviewer comments, and making revisions. Participants also learned how to create a coherent and engaging story for their manuscripts and organize the scientific content using their data and reports.

The workshop was conducted in Spanish and adopted a blend of interactive sessions, presentations, and group activities to engage participants and facilitate learning. Experienced researchers and publishing professionals led sessions on scientific writing techniques, manuscript organization, and effective communication in English. Practical exercises and real-world examples were provided to allow participants to apply the acquired knowledge in a practical setting. Furthermore, participants were provided with comprehensive online resources to facilitate continuous learning beyond the workshop environment.

## Surveys

A total of three surveys were conducted throughout the workshop process: one during registration for interested participants, one prior to the workshop, and one after the workshop.

**Registration survey.**   To facilitate the selection process, participants were asked to complete an open-ended survey during the registration process. This survey aimed to gather demographic information, educational background, professional profiles, and current areas of research among the applicants. The survey aimed to gather insights from participants on three key aspects: their beliefs regarding the reasons for lower publication rates in the Tropical Andes compared to other regions, their main personal limitations to publishing scientific articles, and their suggested strategies for overcoming these challenges. The data collected from this survey played a crucial role in refining and selecting the final participants for the workshop.

**Pre-workshop survey.**   After the final participants were selected, a pre-workshop survey was sent to the chosen participants before the workshop. The survey aimed to gather specific insights from the selected participants, focusing on their areas of interest and the key barriers they aimed to overcome in the workshop related to scientific article writing. Participants were asked to identify their top two sessions of interest, to enable tailoring of the workshop content to match their preferences. Additionally, the survey inquired about the most significant barrier participants wanted to address, providing valuable input to address their specific challenges during the workshop. Lastly, participants were asked about their preferred topics when writing scientific articles so we could better align the workshop content with their research interests. This pre-workshop survey played a crucial role in customizing the workshop experience, ensuring the content was relevant, engaging, and directly addressed the needs and preferences of participants.

**Post-workshop survey.**   Following the workshop, a final survey was conducted to evaluate its impact and gather valuable feedback for future improvements. Participants were asked to rate their overall satisfaction with the workshop and indicate if it met their expectations. Additionally, the survey inquired about the topics related to overcoming obstacles that participants found most valuable, providing insights into the areas that resonated the most with them. Participants were also asked to identify the main session where they learned the most information, allowing organizers to understand the most impactful aspects of the workshop. Moreover, the

survey sought suggestions for specific topics participants would like to be covered in more detail in future hands-on workshops, enabling the organizers to tailor the content to meet the participants' interests and needs. This survey played a crucial role in gauging the workshop's effectiveness and identifying areas for enhancement, ultimately contributing to the continuous improvement of future workshops.

# Results

## Registration survey

**Demographics.** During the five-day open registration period, a total of 520 individuals, mostly from the Tropical Andes region, expressed interest and registered for the workshop. The majority of respondents were from Peru (62.9%), followed by Ecuador (13.8%), Colombia (10.6%), Bolivia (7.1%), and other countries (5.6%). These participants were primarily professionals (85.8%) working in various scientific fields related to natural sciences, including biology, environmental engineering, forestry, agronomy, ecotourism, anthropology, geography, environmental management, and academia, among others. Additionally, 6.9% were currently pursuing undergraduate degrees, and 7.3% were enrolled in graduate programs. The respondents had an average age of 30.39 years, ranging from 19 to 62 years old. In terms of gender distribution, women (56.2%) slightly outnumbered men (43.8%). The median year of undergraduate degree completion was 2010, ranging from 1995 to 2023, while for graduate degrees, it was 2017, ranging from 1990 to 2023. Their research interests covered a wide array of topics, such as biodiversity conservation management, climate change and environmental impact, ecotourism and sustainable tourism, mining and industry, geospatial analysis, remote sensing, herpetology, wildlife, and other closely related fields of study. Additionally, a substantial percentage (67.3%) stated they currently had data or material ready for publishing a scientific manuscript.

**Factors impacting publication rates.** The survey respondents identified several key factors they believe contribute to the lower publication rates of scientific articles in the Tropical Andes compared to other neighboring regions (Fig 1). Commonly highlighted challenges include insufficient training on the publishing process, a cultural emphasis on economic development over research investments, and language barriers. Additionally, respondents cited limited access to research resources, a lack of institutional and governmental support, the high cost of publishing, time limitations, and financial constraints. The survey also revealed challenges related to lower cultural importance on publishing, inadequate guidance in publishing processes, limited awareness of the publication process, restricted investment in research, limited access to indexed journals, and limited career opportunities were also significant factors hindering researchers in the Tropical Andes from publishing their work.

**Personal limitations to publishing.** Survey respondents highlighted eight main limitations that they encounter in their efforts to publish scientific articles (Fig 2). These limitations include insufficient knowledge and experience in the publication process, time constraints due to busy schedules and work commitments, and financial limitations due to high publication costs and limited funding. Language barriers in composing articles in English were also mentioned, along with limited access to essential research resources. Many respondents also conveyed a general lack of self-confidence in their writing skills and expressed fear of rejection from journals. Additionally, respondents expressed difficulties in navigating the complexities of the publication process and manuscript preparation. Lastly, limited collaboration and networking opportunities with peers and mentors were also reported as hurdles to publishing.

**Overcoming challenges.** Survey respondents identified various solutions to address challenges in publishing scientific articles in the region, which can be grouped into four main

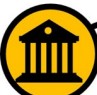 **Limited Support and Training**

Researchers in the region often lack sufficient support and guidance on publishing their research and navigating the publication process

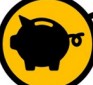 **Funding and Resource constraints**

The region tends to prioritize economic development over scientific research and publications compared to other neighboring regions

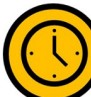 **Language Barriers**

The predominance of English in science poses challenges for researchers since many do not speak or write in English

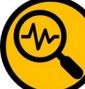 **Lack of Institutional Support**

Limited incentives and support from institutions and governments for publishing

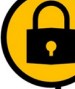 **Publishing costs**

High costs associated with publishing in many journals pose a significant financial challenge

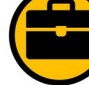 **Time Limitations**

Balancing family commitments with teaching and other work responsibilities restricts time for writing and publishing manuscripts

**Less Emphasis on Publishing**

There is a limited understanding of the publication process and many may not understand the importance of publishing their work

**Restricted Access to Resources**

Local researchers often face difficulties accessing journals, databases, and resources that are not openly available

**Limited Job Opportunities**

Lack of stable career prospects discourage many individuals from investing time and resources into publishing research

**Fig 1. Main factors for lower publication rates in the Tropical Andes.** The most commonly cited reasons by survey respondents in an open-ended questionnaire for the lower publication rates in the Tropical Andes region compared to neighboring countries and other regions.

themes (Fig 3). The first commonly cited strategy focused on training and skill development, highlighting the need for education on the publication process, regular workshops on scientific writing, and specialized programs to help improve individual expertise and self-confidence. The importance of collaborative support networks was also emphasized, involving initiatives such as team-based research, interdisciplinary cooperation, and institutional support systems to foster a nurturing environment. The third common theme centered on resource accessibility, including increasing access to literature and tools, additional funding support, promoting

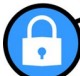
**Insufficient Knowledge and Experience**
Lack of guidance, mentoring, and training on the publication process, leading to uncertainty about effectively navigating the publication process

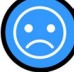
**Time Constraints**
Busy schedules, work commitments, and academic obligations limited amount of adequate time to manuscript writing and preparation

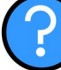
**Financial constraints**
The high publication costs and a limited of research funding creates economic barriers to publishing

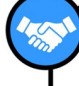
**Language Barriers**
Writing scientific articles in English is challenging due to insufficient language proficiency and the need to improve academic writing skills

**Limited Access to Resources**
Restricted access to research resources such as information, data, journals, and publication platforms

**Lack of Confidence**
Concerns about writing abilities and fear of journal rejection, indicating a need for support to improve skills and confidence

**Manuscript Preparation**
Difficulties organizing ideas, structuring and formatting articles, and understanding publication nuances

**Limited Networking Opportunities**
Lack of collaborative opportunities with peers and mentors hinders research progress and obtaining support for publishing

**Fig 2. Personal barriers to scientific publishing in the Tropical Andes.** The most common personal limitations to publishing scientific articles, as indicated by survey respondents in an open-ended questionnaire.

open science practices, and providing time management and productivity improvement tools. Finally, respondents underscored the need for institutional and cultural transformation, such as cultivating a research culture that rewards scholarly contributions, providing recognition and career advancement incentives, and advocating for policy changes to prioritize and support publishing activities.

## Pre-workshop survey

The pre-workshop survey results revealed several key findings regarding the interests and barriers of the workshop participants in relation to publishing. The two main workshop topics that interested respondents the most for the workshop were "How to transform your data and reports into a compelling story" (44.6%), followed by "Understanding the publication process" (29.3%), and "Overcoming obstacles related to publication" (23.7%) (Fig 4A). When asked what was the single most important barrier to publishing that the participants wished to overcome, nearly a third of respondents chose: "How the publication process works" (33%), followed by "Publishing costs" (18.2%)" and "Access to references" (13.6%) (Fig 4B). Lastly, when asked what specific topic they were most interested in learning, the areas that attracted the greatest interest from participants were "Organizing Information" (37.8%), "Selecting the Appropriate Journal" (31.38%), and again, "Understanding the Publication Process" (29.3%) (Fig 4C).

## Post-workshop survey

The post-workshop survey results revealed a high level of participant satisfaction. A majority of respondents (93.5%) rated the workshop favorably, with 58.7% giving it the highest rating of

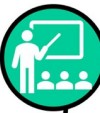

### Training and Skill Development

- Education on the publication process, starting at the undergraduate level
- Regular workshops and training on scientific writing to help researchers stay updated on their skills
- Specialized programs to improve specific expertise and knowledge
- Establish mentorship initiatives to guide researchers in navigating the publication process

### Collaborative Networks and Supportive Environments

- Foster a culture of team-based research and peer support
- Facilitate interdisciplinary cooperation and knowledge sharing
- Provide mentorship and guidance to build confidence and overcome fears
- Develop institutional support systems that provide resources, guidance, and feedback

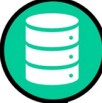

### Resource Accessibility

- Increase access to scientific literature, data, writing tools, and publishing resources
- Secure funding opportunities through grants, scholarships, and research awards
- Implement time management and productivity-enhancing tools
- Promote open science practices to facilitate knowledge exchange

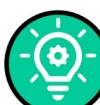

### Institutional and Cultural Transformation

- Cultivate a research culture that prioritizes and rewards scholarly contributions
- Incentivize publication through recognition, awards, and career advancement
- Implement policies and support mechanisms to enable publishing
- Advocate for systemic changes in research practices

**Fig 3. Strategies to overcome publishing challenges in the Tropical Andes.** The four key themes of the proposed strategies were suggested by survey respondents in an open-ended questionnaire to overcome obstacles to publishing scientific articles in the Tropical Andes region.

5. Additionally, an overwhelming majority (93.5%) confirmed that the workshop met their expectations. These findings demonstrate the overall success of the workshop and highlight its positive impact on participants. Among the respondents, 48.9% found the session on understanding the publishing process to be the most valuable, followed by 22.2% who gained valuable information from the session on transforming data and reports into a compelling story (Fig 5A). In terms of overcoming obstacles, the topics that were found most valuable by participants were understanding how the publishing process works (46.6%) and writing in English (27.6%) (Fig 5B). When asked about specific topics they would like to focus on in a future hands-on workshop, participants expressed interest in various areas. The topics with the highest preferences were how to organize information (20.8%), data visualization (14.6%), and selecting the right journal (15.6%) (Fig 5C).

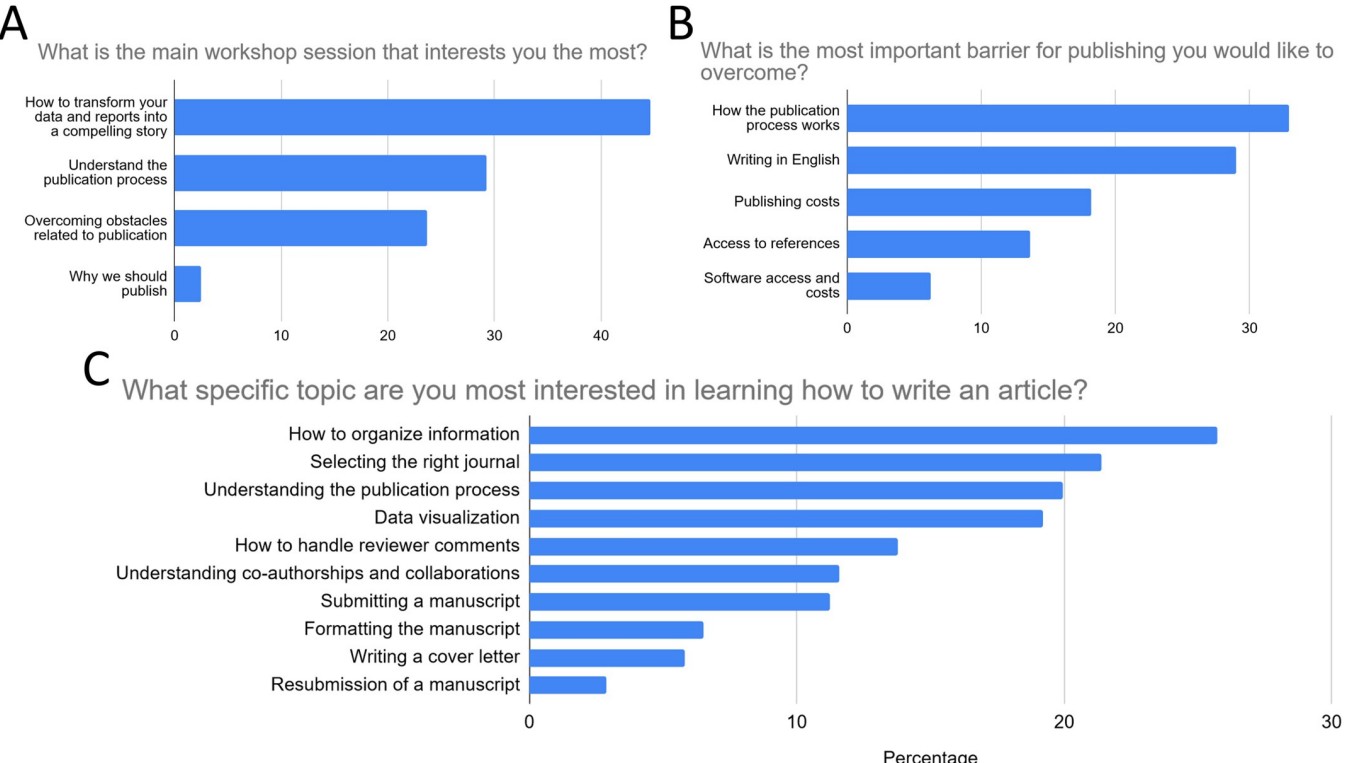

**Fig 4. Key findings from the pre-workshop survey.** The figures represent the total percentage of responses for (A) the top two general workshop topics participants were interested in, (B) the main barrier to publishing participants wanted to overcome, and (C) the specific writing and publishing topic of most interest for participants.

## Discussion

The survey results, combined with insights from the workshops, provide a comprehensive understanding of the multifaceted challenges researchers in the Tropical Andes encounter when publishing their scientific work. These challenges are deeply embedded within the region's research landscape, as evidenced by the diverse backgrounds of workshop participants and the significant interest garnered from across Latin America, particularly Peru, Ecuador, Bolivia, and Colombia. Notably, the interested participants were predominantly experienced professionals with over a decade of expertise and advanced degrees, rather than early-career researchers or students. This unique composition underscores that publishing barriers do not stem from a lack of experience or qualifications, but rather from systemic issues ingrained within the research infrastructure of the region, such as limited access to publishing resources and institutional support. Additionally, nearly two-thirds of participants indicated having datasets and research materials ready for publication, further revealing the large reservoir of untapped and unrealized scientific knowledge in the region. Addressing these regional challenges is crucial for empowering local researchers, sharing valuable contributions with the broader scientific community, and unlocking the full potential of scientific research in the Tropical Andes.

### Factors impacting the publication divide

The survey respondents highlight several critical factors they believe contribute to the lower publication rates of researchers in the Tropical Andes region compared to other regions.

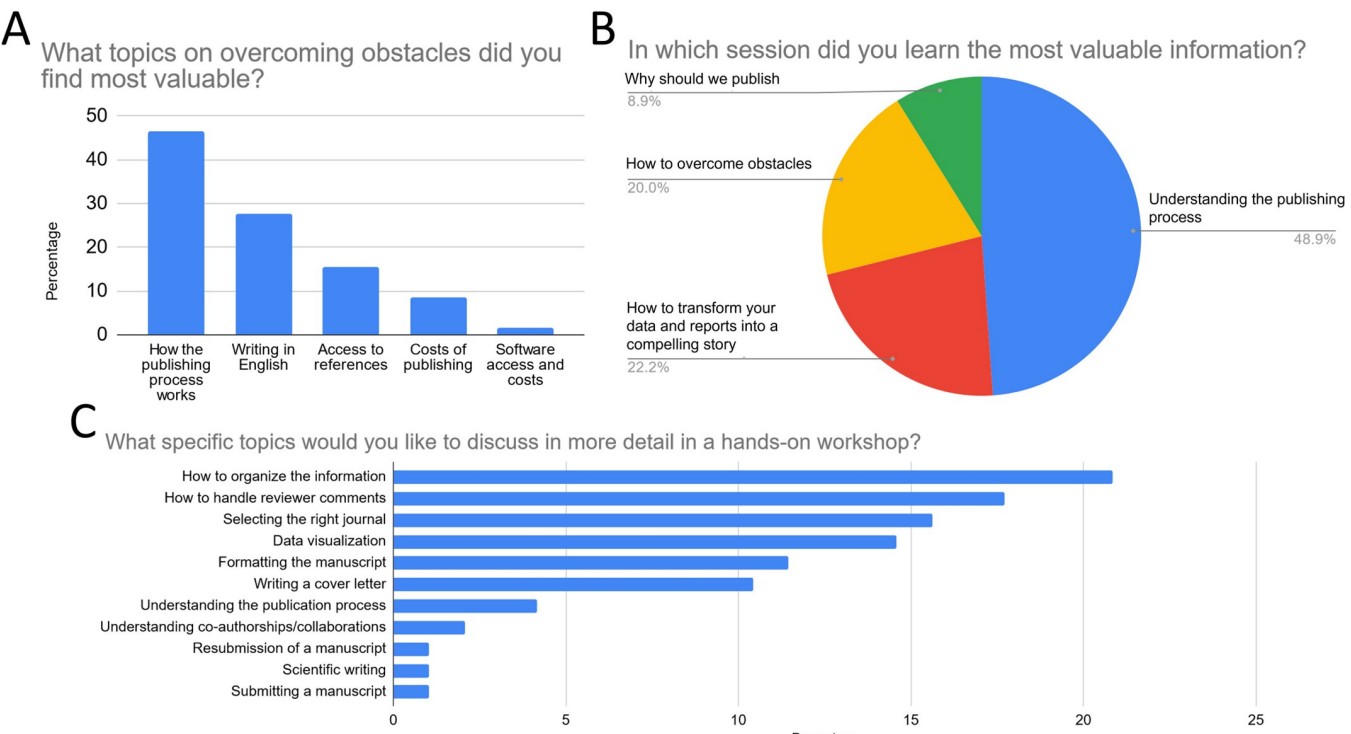

**Fig 5. Key findings from the post-workshop survey.** The figures represent the total percentage of responses for (A) the general workshop topic participants found most valuable, (B) the workshop session participants learned the most from, and (C) the specific writing and publishing topic participants were most interested in for in-depth future workshops.

These challenges are varied, yet interconnected, reflecting the complexity of the scientific landscape in the region. Notably, respondents highlighted a lack of sufficient training and institutional support structures for navigating the publishing process as a significant factor for the publication gap. The lack of guidance and support can leave researchers feeling overwhelmed and ill-equipped to navigate the intricacies of scholarly publishing, ultimately discouraging them from publishing their valuable work. Another commonly cited reason was the lack of funding and resources available for scientific research in the region. This is not all surprising since countries in the Tropical Andes, such as Bolivia, Ecuador, and Peru, often have smaller economies and allocate far less public funding to research and publishing compared to the global average [5, 6, 24]. However, this lack of investment severely limits the ability of researchers not only to publish in high-impact journals due to the high costs, but also the ability to access many articles, journals, databases, and other resources that are not openly available [25, 26]. Additionally, the lack of stable, long-term research funding in the region frequently forces scientists to rely on sporadic, short-term grants, further limiting the scope and depth of their investigations [27, 28].

Survey respondents also highlighted the challenge of publishing in English-language journals, which is particularly relevant to the Tropical Andes given their relatively lower English proficiency compared to even other Latin American countries [29]. This language barrier significantly increases the time and effort required for non-native English speakers to prepare manuscripts and navigate the publication process, hindering their active engagement in international scientific discussions and undermining their professional growth and recognition within the scientific community [30, 31]. However, despite the large number of open-access Spanish language journals and the comparable growth of these articles to English publications

[32], weaker incentives and limited institutional pressure to publish were also cited as factors discouraging researchers from disseminating their work. In many of these countries, publishing in academic journals may not be as highly prioritized for career advancement, as the incentive structure often lacks a strong emphasis on publications [21, 33]. Researchers often focus more on local impact and engagement, such as reports for policy and decision-makers, or addressing immediate community needs, rather than pursuing recognition through scientific articles [21]. Consequently, due to a lack of institutional and cultural incentives, many researchers may not understand or appreciate the importance of publishing for the broader scientific community, as also demonstrated by the survey findings. Furthermore, the lack of stable career prospects and time constraints, compounded by balancing family and work commitments, were also stated as possible factors deterring many researchers from investing time and resources into publishing their research. This research landscape contrasts sharply with more prolific publishing nations or regions, which typically benefit from a more established scientific culture, greater financial resources, and stronger research infrastructure and support systems focused on publication.

## Personal barriers

The survey respondents highlighted a range of personal limitations that parallel the broader regional challenges. A common barrier that surfaced from the open-ended questionnaire was an inadequate understanding and experience in the publication process. The pre-workshop survey results corroborate these findings, with a third of respondents identifying "How the publication process works" as the single most important barrier they wished to overcome, even outweighing concerns about costs and language proficiency. Importantly, the lack of understanding can directly impact self-confidence and motivation to publish, as evidenced by the registration survey responses which revealed widespread self-doubt regarding writing skills, fear of journal rejections, and difficulties in navigating the manuscript preparation process. These results reveal a fundamental knowledge gap that can directly impact their self-confidence and motivation to pursue publication. This is evident in the survey responses which highlighted widespread self-doubt about their writing abilities, fear of journal rejections, and difficulties in navigating the complexities of manuscript preparation. As previously stated, most researchers expressed that they had work ready for publication, but their lack of confidence in their capabilities and the research itself deterred them from submitting their work. Considering most participants are already professionals in their field, this finding highlights the transformative potential of simply demystifying the nuances of the publication process. By providing researchers with a clearer understanding of the process - such as navigating manuscript preparation, adhering to journal guidelines, addressing reviewer comments, and understanding the intricacies of publishing - we can boost self-confidence and empower these local experts to share their research, ultimately transforming the publishing landscape in the Tropical Andes.

However, personal knowledge limitations were also compounded by broader systemic issues. Financial barriers were cited as another major personal obstacle, with respondents highlighting the high prohibitive costs associated with publishing in high-impact, international journals as a significant barrier. This issue is particularly acute given the continuing underfunding of research in the Tropical Andes, which constrains individual researchers' access to the resources and institutional support needed to offset these publication expenses [9, 16, 19, 25, 26]. Time constraints also played a significant role, as individuals stated they struggled to balance scientific work with other professional and personal commitments, limiting the time and focus to publication efforts. Language barriers in composing articles in English further

exacerbate these hurdles, as many researchers in the region do not feel proficient or confident enough to effectively communicate their findings to the broader global scientific community. Finally, limited opportunities for collaboration and networking with peers and mentors emerged as yet another personal barrier, hindering the ability to exchange ideas, seek support, and foster partnerships critical for advancing their work. Addressing the multifaceted limitations identified by local researchers is essential to unlock the full research potential of the Tropical Andes and amplify their valuable contributions to the broader scientific community.

## Solutions to challenges

The survey respondents highlighted a range of potential strategies to address the personal and systemic barriers that limit their ability to successfully publish and disseminate their research. These proposed solutions provide valuable insights into the specific needs and preferences of the Tropical Andes research community.

**Training and skill development.** The most commonly cited strategy focused on the need for targeted training and skill development, emphasizing the critical need for building and strengthening individual capacities related to the publication process. Respondents emphasized the importance of education in the publication process, starting at the undergraduate level, to familiarize researchers with the expectations and norms of scholarly publishing. Regular workshops and specialized training sessions on academic writing, scientific communication, and programs aimed at enhancing expertise and self-confidence are key to empowering researchers in navigating the complex publication landscape more effectively. These initiatives help researchers refine their skills, develop effective strategies for submitting and publishing high-quality manuscripts, and stay up-to-date on best practices of best practices in scientific writing [34–36]. This strategy has been found to not only improve researcher expertise but also self-confidence, which is critical for individual professional growth [37, 38]. Establishing mentorship initiatives that pair early-career or novice researchers with experienced scholars can also be a valuable strategy to guide them in navigating the complex publication process and further develop the necessary skills and confidence to write compelling narratives, effectively convey their research findings, and successfully navigate the peer-review process [35, 39].

**Collaborative networks and supportive environments.** Another crucial aspect emphasized by the respondents is the importance of collaborative research networks and supportive environments. Initiatives such as team-based research, interdisciplinary cooperation, and institutional support systems can facilitate the exchange of knowledge, mutual learning, provide mentorship opportunities, and create a sense of community among researchers. Such collaborative endeavors not only enhance the quality of research outputs, leading to higher publication rates, greater research impact, and the emergence of innovative problem-solving approaches, but also offer valuable opportunities for researchers to learn from their peers, gain new perspectives, and develop their professional networks [40–42]. Expanding scientific networks through international collaboration, particularly with well-resourced institutions, provides researchers from underrepresented or resource-constrained areas, such as those in the Tropical Andes, with access to a broader pool of expertise, perspectives, and resources [43–45]. Moreover, fostering collaborations with English-speaking researchers can facilitate language skill development and provide valuable opportunities for knowledge exchange and mentorship [44, 45]. Developing institutional support systems that also offer resources, guidance, and constructive feedback, whether through formal programs or informal support systems, can further empower researchers to thrive and navigate the publication landscape successfully [37].

**Resource accessibility.** The survey results also focused on addressing resource accessibility, recognizing the importance of addressing the challenges researchers face in accessing

essential resources for writing, publication, and dissemination. By providing wider access to essential resources like relevant literature, data, writing tools, and publishing platforms, researchers can significantly improve the quality of their research and ultimately increase their chances of successful publication. However, given the resource constraints often faced by researchers in the Tropical Andes region, it may not always be possible to significantly increase the available resources. In such cases, researchers can leverage a variety of free and cost-effective tools to optimize their workflow and maximize output despite limited resources. For example, they can use free reference management software (like Zotero, Mendeley) to organize literature, and open-source statistical software (such as R, Python) for data analysis, reducing the need for expensive proprietary tools. Collaborating through free cloud-based platforms (Google Drive, Dropbox) can facilitate resource sharing and manuscript co-authoring. Free online courses and webinars (Coursera, edX, Udemy, YouTube) can enhance research skills and stay current with methodologies. Additionally, language translation tools (Google Translate, DeepL) can help overcome language barriers, while AI-powered writing assistants (ChatGPT, other large language models) can provide support for tasks such as writing, editing, and proofreading [45, 46]. Participating in online research communities and forums (ResearchGate, Twitter, Facebook groups) allows idea exchange and network building without travel. Submitting preprints to free repositories (arXiv, bioRxiv) enables quick dissemination and feedback prior to publication [47]. Prioritizing open-access journals and negotiating for fee waivers can ensure wider accessibility. Lastly, applying for targeted small grants, engaging in community-based participatory research, and building local partnerships can further provide financial support, leverage existing resources, and ensure research-relevant impact.

**Institutional and cultural transformation.** The survey respondents emphasized the need for an institutional and cultural transformation that values and rewards scholarly contributions. Mechanisms such as recognition, awards, and career advancement incentives were highlighted as effective ways to encourage researchers to engage in publishing activities. However, this does not mean simply adhering to the typical "publish or perish" culture that can lead to problematic incentives and practices. Rather, the respondents highlighted the need for a more nuanced, balanced, and thoughtful approach to valuing and rewarding scholarly contributions. These incentives should be carefully designed to motivate meaningful, high-quality publications that advance scientific understanding and have tangible societal impact, rather than simply rewarding publication volume. Additionally, the responses underscored the need for policies and support systems that enable researchers to dedicate adequate time and resources to publishing. This includes provisions for flexible schedules and protected time for research, writing, and professional development workshops. Moreover, a general shift in the research culture is needed, given that many respondents stated they didn't know or understand why they should publish. This is particularly crucial considering that many researchers in the region are currently either unwilling or unable to openly share their data or findings due to various factors such as concerns about intellectual property rights, fears of misuse, and general lack of awareness about the benefits of sharing with the broader scientific community [21, 48]. Such transformative shifts through carefully designed incentives, policies, and support systems, could significantly expand the body of scientific research from the region. For example, while Ecuador's scientific production has historically lagged behind other Latin American countries due to limited research culture and prioritization, recent government policies aimed at addressing this issue have led to remarkable growth in research publications, outpacing the Latin American average [23]. By cultivating a research culture that values and supports scholarly contributions, institutions can drive a lasting transformative cultural and institutional shift that aligns with the region's global significance.

## Workshop feedback

The post-workshop survey results highlighted the significant impact and success of the three-day workshop, demonstrating high participant satisfaction and emphasizing its positive contribution to researchers in the region. A substantial majority rated the workshop favorably and stated that it met their expectations. Participants particularly valued the sessions on "understanding the publication process" and "transforming data into compelling narratives", underscoring the importance of equipping researchers in the region with understanding the publication process and improving their communication skills. Importantly, the workshop also aligned with the solutions proposed by the participants themselves, which emphasized continuous learning and improvement, skill development, supportive environments, fostering confidence and motivation, providing access to resources and information, cultivating a research-focused culture, and ongoing education and training. The workshop's effectiveness in addressing researchers' needs and empowering them to navigate the publication process and communicate their findings underscores the importance of such initiatives.

## Limitations

While this study offers valuable insights into the challenges faced by individuals in the Tropical Andes region regarding scientific publishing, several limitations should be acknowledged. First, the registration survey was not originally designed for research purposes, but rather as a tool for selecting participants and designing the workshop. The open-ended nature of the questions required the data to be retrospectively categorized and coded, which may have introduced subjectivity. Additionally, while over half of the participants were selected for the workshop, the decision to maintain anonymity meant there was no record of attendance or a way to link individual responses across the pre- and post-workshop surveys. Many participants could only attend for one day or leave before completing the surveys, further limiting the ability to capture complete data. This lack of identifying information also prevented the exploration of potential differences in challenges based on factors like nationality, institution type, or career stage. Furthermore, the qualitative approach restricts the ability to statistically compare the findings with other studies or generalize the results to a broader population of the Tropical Andes. However, this approach aligns with the primary aim of gathering rich, contextual data from the targeted population of researchers in the Tropical Andes region who were actively seeking support for publishing their work. The intent of this study was to understand their specific perspectives and needs in overcoming barriers, rather than producing statistically representative findings. Despite these limitations, the study provides valuable insights that can inform future research and targeted initiatives to address the publishing challenges faced by this subset of the research community. By focusing on the perspectives of those actively seeking solutions, the findings shed light on the specific needs and obstacles that could be prioritized in efforts to strengthen the region's scientific publishing capacity and support.

## Future directions

Building on the insights from this study, we can foster a thriving research culture in the Tropical Andes by addressing identified challenges and implementing proposed solutions. Future initiatives should focus on equipping researchers with the necessary tools, skills, and support to effectively navigate the publication process. This includes offering targeted training programs covering essential topics such as information organization, data visualization, and journal selection–areas highlighted by participants as priorities for future workshops. The positive feedback received from the workshop demonstrates the value of such initiatives, which,

alongside ongoing training, institutional support, mentorship, and networking opportunities, can empower this critically important research community by providing them with tools to overcome the challenges they face and narrow the publication divide. Future research can build upon these insights, evaluating the impact of various interventions and strategies on publication rates and further exploring how institutional and governmental support can contribute to further improving the publishing landscape.

## Conclusions

The limited visibility and dissemination of research findings from the region can lead to a lack of recognition for the valuable contributions made by local researchers, perpetuating a cycle of reduced funding opportunities, limited career advancement prospects, and diminished incentives for research and publishing activities [12, 22, 49]. Moreover, the underrepresentation of research from the Tropical Andes in the global scientific discourse can skew understanding of the region's unique ecological, social, and economic contexts, hindering the development of effective solutions to pressing challenges [50]. Addressing the factors impacting the publication divide is not only crucial for empowering local researchers but also for ensuring that the knowledge generated in the Tropical Andes is effectively integrated into the global scientific community, fostering a more comprehensive and inclusive understanding of the world's biodiversity and ecosystems. By identifying and addressing specific regional needs, we can bridge the gap between knowledge creation and dissemination, empowering all researchers and unlocking the full potential of the global research community.

## Positionality statement

As the authors of this article, our positions are deeply rooted in both the Latin American region and the specific context of the Tropical Andes. With three team members originating from the region, we possess a personal understanding of the socio-economic, cultural, and environmental factors that shape ecological research within the Tropical Andes. Two authors still reside and actively engage in research within the Tropical Andes, providing us with first-hand insights into the challenges researchers face, the complexities of biodiversity conservation, and the interactions between scientific research and local communities. This combination of backgrounds and experiences informs our perspectives and frames our approach to navigating the intricacies of the publication gap and ecological research within this ecologically significant region, offering comprehensive insights and context-specific solutions that stem from our intimate engagement with the diverse landscapes, communities, and challenges of the Tropical Andes.

## Supporting information

**S1 Appendix. Workshop promotion and itinerary.** Promotional materials and detailed itinerary for the 3-day scientific writing and publishing workshop.
(PDF)

**S1 Data. Survey responses.** This file contains the raw anonymized data collected from the registration, pre-workshop, and post-workshop survey. It includes participant responses and survey questions.
(XLSX)

## Author Contributions

**Conceptualization:** Jose W. Valdez.

**Data curation:** Jose W. Valdez, Lucía Castro Vergara.

**Formal analysis:** Jose W. Valdez.

**Funding acquisition:** Miguel Fernandez.

**Investigation:** Jose W. Valdez.

**Methodology:** Jose W. Valdez, Lucía Castro Vergara.

**Project administration:** Lucía Castro Vergara, Gabriela Orihuela, Miguel Fernandez.

**Resources:** Miguel Fernandez.

**Supervision:** Jose W. Valdez, Miguel Fernandez.

**Visualization:** Jose W. Valdez, Lucía Castro Vergara.

**Writing – original draft:** Jose W. Valdez, Lucía Castro Vergara, Miguel Fernandez.

**Writing – review & editing:** Jose W. Valdez, Lucía Castro Vergara, Gabriela Orihuela, Miguel Fernandez.

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
