## [Decision Letter · Decision Letter 0]

30 Apr 2024

PONE-D-24-01605Understanding the Biodiversity Research Publication Gap in the Tropical Andes: Unveiling Challenges and Implementing SolutionsPLOS ONE

Dear Dr. Valdez,

Thank you for submitting your manuscript to PLOS ONE. After careful consideration, we feel that it has merit but does not fully meet PLOS ONE’s publication criteria as it currently stands. Therefore, we invite you to submit a revised version of the manuscript that addresses the points raised during the review process.

 I am pleased to advise you of the decision of PLOS ONE to accept your manuscript, pending edits as indicated by the reviewers below.

In addressing the reviewers' comments, I suggest to add a sub-section of your paper, in the Methods section, named "Limitations of the study".

In this sub-section, you could elaborate a bit on the negative comments that the reviewers provided. For example, I encourage you to take advantage of this section to explain limitations picked up by either reviewer towards the points below for which the reviewer considered that 'No', some publication criteria were not met.

This would be an opportunity for you to shed some light on why the paper is sound even if some flaws might be detected by your audience.

Also, please consider carefully and address comments highlighted at point 5 below, by each reviewer. Please ensure that your decision is justified on PLOS ONE’s publication criteria and not, for example, on novelty or perceived impact.

We look forward to receiving your revised manuscript.

Kind regards,

Umberto Baresi, Ph.D.

Academic Editor

PLOS ONE

 [This research received funding from the ERANet Joint Call 2016–2017 (DLR Förderkennzeichen 01DN19032 Tropical Andes Observatory—TAO).].  

5. We notice that your supplementary figures are included in the manuscript file. Please remove them and upload them with the file type 'Supporting Information'. Please ensure that each Supporting Information file has a legend listed in the manuscript after the references list.

7. We note that Supplementary 1 in your submission contain copyrighted images. All PLOS content is published under the Creative Commons Attribution License (CC BY 4.0), which means that the manuscript, images, and Supporting Information files will be freely available online, and any third party is permitted to access, download, copy, distribute, and use these materials in any way, even commercially, with proper attribution. For more information, see our copyright guidelines: http://journals.plos.org/plosone/s/licenses-and-copyright.

a. You may seek permission from the original copyright holder of Supplementary 1 to publish the content specifically under the CC BY 4.0 license. 

Additional Editor Comments :

Dear author,

I am pleased to advise you of the decision of PLOS ONE to accept your manuscript, pending edits as indicated by the reviewers below.

In addressing the reviewers' comments, I suggest to add a sub-section of your paper, in the Methods section, named "Limitations of the study".

In this sub-section, you could elaborate a bit on the negative comments that the reviewers provided. For example, I encourage you to take advantage of this section to explain limitations picked up by either reviewer towards the points below for which the reviewer considered that 'No', some publication criteria were not met.

This would be an opportunity for you to shed some light on why the paper is sound even if some flaws might be detected by your audience.

Also, please consider carefully and address comments highlighted at point 5 below, by each reviewer.

I look forward to receiving the revised version of your manuscript.

Best regards,

Umberto Baresi, PhD

Reviewers' comments:

Reviewer's Responses to Questions

**Comments to the Author**

1. Is the manuscript technically sound, and do the data support the conclusions?

Reviewer #1: No

Reviewer #2: Yes

2. Has the statistical analysis been performed appropriately and rigorously? 

Reviewer #1: No

Reviewer #2: No

3. Have the authors made all data underlying the findings in their manuscript fully available?

Reviewer #1: No

Reviewer #2: No

4. Is the manuscript presented in an intelligible fashion and written in standard English?

Reviewer #1: Yes

Reviewer #2: Yes

5. Review Comments to the Author

Reviewer #1: Authors manuscript is well written and in standard English. However there are several flaws on the design of data adquisition. Authors based their results on people registered for a workshop which was advertised on social media. Authors do not compare their results for instance with the data from Peruvian researchers with CTI vitae from CONCYTEC (Consejo Nacional de Ciencia y Tecnología) which is an open source about researcher publications and other information.

Their sample is very small and not random.

There is not any statistical analyses so it is difficult to apply their results to a more general population, much less the tropical Andes

The criteria for their selection of registered researchers is not clear. Authors should have included the raw data as supplementary material. We do not know how many of the selected surveyed people work or are part of a university, national research institution, private research institution or are free researchers.

Reviewer #2: Figure 2 does not specify which subplots correspond to A and B, and the legend of Figure 2B is small and challenging to read.

In Figure 3, we depict the primary findings derived from the survey responses. Given the nature of the data, employing a Pareto chart would be more appropriate for effectively highlighting the main reasons across all topics."

Exploring nationality-based response variations could provide valuable insights. Utilizing a chi-square test for analysis, non-significant results may suggest similarities among the countries, while significant differences would indicate distinctions despite their shared continent.

6. PLOS authors have the option to publish the peer review history of their article (what does this mean?). If published, this will include your full peer review and any attached files.

Reviewer #1: No

Reviewer #2: No

---

## [Author Response · Author response to Decision Letter 0]

6 Jun 2024

RESPONSE TO REVIEWERS

 Dear author,

 I am pleased to advise you of the decision of PLOS ONE to accept your manuscript, pending edits as indicated by the reviewers below.

 In addressing the reviewers' comments, I suggest to add a sub-section of your paper, in the Methods section, named "Limitations of the study".

 In this sub-section, you could elaborate a bit on the negative comments that the reviewers provided. For example, I encourage you to take advantage of this section to explain limitations picked up by either reviewer towards the points below for which the reviewer considered that 'No', some publication criteria were not met.

 This would be an opportunity for you to shed some light on why the paper is sound even if some flaws might be detected by your audience.

 Also, please consider carefully and address comments highlighted at point 5 below, by each reviewer.

 I look forward to receiving the revised version of your manuscript.

 Best regards,

 Umberto Baresi, PhD

RESPONSE: Thank you for informing us of PLOS ONE's decision to accept our manuscript, pending the revisions suggested by the reviewers. We are grateful for the opportunity to address their constructive feedback.

In response to your recommendations, we have added a new sub-section titled "Limitations" within the Discussion section of the manuscript. This allowed us to thoughtfully address any limitations or concerns raised by the reviewers, while also making it clear why the overall study design and findings remain sound. We believe this addition will provide valuable context for the readers. Furthermore, we have carefully considered and incorporated the comments highlighted at point 5 by each reviewer. This has resulted in several improvements throughout the manuscript. Notably, we have revised the tables and converted them to figures to enhance clarity and visual impact. Additionally, we made other minor edits to improve the overall quality and coherence of the work, including thoughtfully refining the language to ensure a more inclusive tone. We have also updated the title of the paper to "Overcoming the Tropical Andes Publication Divide: Insights from Local Researchers on Challenges and Solutions." This new title better reflects the qualitative nature of our study and its focus on providing in-depth insights from researchers in lower publication countries in the Tropical Andes. We appreciate your insightful feedback and believe these changes significantly enhance the manuscript. Please let us know if there are any other areas that require further adjustment. Specific responses to reviewers comments are below. 

 Reviewers' comments:

 Reviewer's Responses to Questions

Comments to the Author

 1. Is the manuscript technically sound, and do the data support the conclusions?

Reviewer #1: No

Reviewer #2: Yes

RESPONSE: We have addressed reviewer 1’s specific comments on this issue below.

2. Has the statistical analysis been performed appropriately and rigorously?

Reviewer #1: No

Reviewer #2: No

RESPONSE: We have not performed any statistical analyses as it is a quantitative study. We have addressed this in greater detail below.________________________________________

3. Have the authors made all data underlying the findings in their manuscript fully available?

Reviewer #1: No

Reviewer #2: No

RESPONSE: We have now included the anonymized dataset as supplementary material.

4. Is the manuscript presented in an intelligible fashion and written in standard English?

Reviewer #1: Yes

Reviewer #2: Yes

NO RESPONSE NEEDED.

5. Review Comments to the Author

Reviewer #1: Authors manuscript is well written and in standard English. However there are several flaws on the design of data adquisition. Authors based their results on people registered for a workshop which was advertised on social media. Authors do not compare their results for instance with the data from Peruvian researchers with CTI vitae from CONCYTEC (Consejo Nacional de Ciencia y Tecnología) which is an open source about researcher publications and other information.

RESPONSE: Thank you for your thoughtful comments and critique of our study. We appreciate the opportunity to address your concerns. While we acknowledge that this approach may introduce potential biases, it aligns with the specific objectives of our study. Our primary goal was not to obtain a strictly representative sample of the entire research community in the Tropical Andes region. Instead, we aimed to gather insights from researchers who were actively interested in and motivated to address the challenges associated with scientific publishing. The overwhelming response rate, with over 500 applicants expressing interest within a short period, demonstrates a substantial pool of researchers facing similar challenges and a widespread interest in the topic. We have now added this to the manuscript to make it clearer “The primary aim was to gather in-depth perspectives from those actively seeking support and guidance to overcome challenges in publishing their research.” Additionally, while comparing our results with databases such as CONCYTEC could provide additional perspectives, it was not within the scope or purpose of our study. Our research focused on gathering firsthand experiences and perspectives directly from researchers in the region through qualitative methods. Furthermore, the CONCYTEC database is specific to Peru, whereas our study aimed to capture insights from multiple countries within the Tropical Andes region, including Ecuador and Bolivia. However, we have now added a "Limitations of the Study" section in the Discussion which elaborates on this and other limitations.

 Their sample is very small and not random.

RESPONSE: We disagree with the characterization of our sample as very small. Our study included over 500 respondents, which provides a substantial dataset for qualitative analysis. However, we acknowledge that the sample is not random. It comprises researchers who expressed interest in a workshop on publishing, thus representing those particularly motivated to address publishing challenges. We have now made this clearer in the manuscript: “The primary aim was to gather in-depth perspectives from those actively seeking support and guidance to overcome challenges in publishing their research.” However, we have now added a "Limitations of the Study" section in the Discussion, as suggested. This section elaborates on the limitations picked up by the reviewers, including the non-random nature of our sample and the lack of statistical analyses, and explains why the study is still valuable despite these limitations.

There is not any statistical analyses so it is difficult to apply their results to a more general population, much less the tropical Andes

RESPONSE: While we agree that our study does not include statistical analyses, this is because it is a qualitative study by design. The focus is on gathering detailed insights and experiences from researchers rather than producing generalizable statistical data. Qualitative research is particularly suited to exploring complex issues in depth, capturing the nuances of participants' experiences, and providing insights into their perspectives. Our approach allowed us to identify specific barriers and needs that researchers in the region face, which might be overlooked in quantitative studies. 

It is also important to clarify that our study does not aim to generalize findings to the entire Andes region. As stated in the introduction, our focus was on Bolivia, Ecuador, and Peru, countries that publish the least in Latin America. We dedicated an entire paragraph mentioning Colombia, which has a significantly higher publication rate, as an outlier in the region. Therefore, our goal was not to generalize to the entire Tropical Andes but to focus on the specific context and challenges within these three countries in particular. 

Nevertheless, we have now changed the title of the paper to "Overcoming the Tropical Andes Publication Divide: Insights from Local Researchers on Challenges and Solutions." This new title better reflects the qualitative nature of our study and its focus on providing in-depth insights from researchers in lower publication countries in the Tropical Andes, rather than attempting to produce generalizable statistical data.

Additionally, we have made this more clear in the limitations section: “...the qualitative approach restricts the ability to statistically compare the findings with other studies or generalize the results to a broader population of the Tropical Andes. However, this approach aligns with the primary aim of gathering rich, contextual data from the targeted population of researchers in the Tropical Andes region who were actively seeking support for publishing their work. The intent of this study was to understand their specific perspectives and needs in overcoming barriers, rather than producing statistically representative findings. Despite these limitations, the study provides valuable insights that can inform future research and targeted initiatives to address the publishing challenges faced by this subset of the research community.”

 The criteria for their selection of registered researchers is not clear. Authors should have included the raw data as supplementary material. We do not know how many of the selected surveyed people work or are part of a university, national research institution, private research institution or are free researchers.

RESPONSE: We acknowledge that our initial explanation regarding the selection criteria for registered researchers may have lacked clarity. Our criteria aimed to assemble a diverse and experienced group of participants from various backgrounds, including academia, national research institutions, private research institutions, and independent researchers. Factors such as geographical origin, previous participation in relevant workshops, academic background, and expertise were carefully considered during the selection process. We have now provided a detailed explanation of our selection process in the Methodology section: “From the pool of over 500 registered applicants, we carefully selected 292 participants to take part in the workshop. The selection process focused on geographical origin (prioritizing the Tropical Andes region, particularly Peru, Bolivia, Ecuador, Colombia, and Venezuela), previous participation in related workshops and those on the ACCA mailing list (indicating relevant interest and engagement), age and academic background (excluding applicants 22 years old or younger, recent undergraduates within the last year and prioritizing those with postgraduate education), academic expertise (favoring ecology and conservation over communication and social sciences), English proficiency, and individuals with material ready for publishing. After thoroughly evaluating survey responses against these criteria, we were able to have a select group of participants with diverse expertise and experiences, enabling us to tailor the workshop to meet their needs and expectations.” We believe this detailed explanation enhances the transparency and understanding of our methodology for participant selection.

Additionally, while we had detailed information from the registration survey, maintaining participant anonymity was fundamental to fostering open and honest responses in our study. However, the inherent flexibility of our study design, allowing participants to freely enter and exit the workshop during the three days, presented challenges in accurately attendance and survey completion. Despite our efforts to ensure a representative sample from the registration demographics, it was not possible to know which of the registered individuals ultimately participated in the workshop, adding complexity to our analysis. This fluidity made it difficult to link workplace data collected during registration to specific participants. Given these constraints, we removed the demography of the selected participants section, which was only two sentences, from the results as we cannot confidently associate this information with the general demography of the individuals who ultimately participated in the workshops. We have made this clear in the limitations section: “While this study offers valuable insights into the challenges faced by individuals in the Tropical Andes region regarding scientific publishing, several limitations should be acknowledged. First, the registration survey was not originally designed for research purposes, but rather as a tool for selecting participants and designing the workshop. The open-ended nature of the questions required the data to be retrospectively categorized and coded, which may have introduced subjectivity. Additionally, while over half of the participants were selected for the workshop, the decision to maintain anonymity meant there is no record of attendance or way to link individual responses across the pre- and post-workshop surveys. Many participants could only attend for one day or leave before completing the surveys, further limiting the ability to capture complete data. This lack of identifying information also prevented exploration of potential differences in challenges based on factors like nationality, institution type, or career stage.” Lastly, in response to the reviewer’s suggestion, we have now added the anonymized survey data as supplementary material, removing any identifying information such as names and emails. This addition enhances the transparency and accessibility of our data.

Reviewer #2: Figure 2 does not specify which subplots correspond to A and B, and the legend of Figure 2B is small and challenging to read.

RESPONSE: We have now removed this figure as explained in the previous comment. 

 In Figure 3, we depict the primary findings derived from the survey responses. Given the nature of the data, employing a Pareto chart would be more appropriate for effectively highlighting the main reasons across all topics."

RESPONSE: The Pareto chart was not used because each participant could provide multiple responses. This would result in a cumulative percentage exceeding 100%, rendering the Pareto chart unsuitable for accurately representing the data.

Exploring nationality-based response variations could provide valuable insights. Utilizing a chi-square test for analysis, non-significant results may suggest similarities among the countries, while significant differences would indicate distinctions despite their shared continent.

RESPONSE: This is a great idea. However, while we initially included nationality questions in the pre-selection survey, we overlooked their inclus

---

## [Editor Report · Decision Letter 1]

9 Jun 2024

PONE-D-24-01605R1Overcoming the Tropical Andes publication divide: insights from local researchers on challenges and solutionsPLOS ONE

Dear Dr. Valdez,

Thank you for submitting your manuscript to PLOS ONE. After careful consideration, we feel that it has merit but does not fully meet PLOS ONE’s publication criteria as it currently stands. Therefore, we invite you to submit a revised version of the manuscript that addresses the points raised during the review process.

We look forward to receiving your revised manuscript.

Kind regards,

Umberto Baresi, Ph.D.

Academic Editor

PLOS ONE

Journal Requirements:

Additional Editor Comments:

I would like to thank the author for the extra effort that they put into this manuscript.

The reviewers' comments have been addressed, and the manuscript is almost ready for publication.

Overall, the manuscript has greatly improved.

There are minor edits required, before the paper can be accepted for publication. These are:

- Check the number of figures, which is now not accurate.

- Check the numbering of sections; the latest version does not show accurate numbering.

Overall, I suggest a general check of the whole paper before resubmitting it in its final version.

Thank you for submitting your research for publication in PLoS ONE.

Kind regards

Umberto Baresi - on behalf of the Editorial Board

---

## [Author Response · Author response to Decision Letter 1]

10 Jun 2024

RESPONSE TO REVIEWERS

 Dear author,

 I am pleased to advise you of the decision of PLOS ONE to accept your manuscript, pending edits as indicated by the reviewers below.

 In addressing the reviewers' comments, I suggest to add a sub-section of your paper, in the Methods section, named "Limitations of the study".

 In this sub-section, you could elaborate a bit on the negative comments that the reviewers provided. For example, I encourage you to take advantage of this section to explain limitations picked up by either reviewer towards the points below for which the reviewer considered that 'No', some publication criteria were not met.

 This would be an opportunity for you to shed some light on why the paper is sound even if some flaws might be detected by your audience.

 Also, please consider carefully and address comments highlighted at point 5 below, by each reviewer.

 I look forward to receiving the revised version of your manuscript.

 Best regards,

 Umberto Baresi, PhD

RESPONSE: Thank you for informing us of PLOS ONE's decision to accept our manuscript, pending the revisions suggested by the reviewers. We are grateful for the opportunity to address their constructive feedback.

In response to your recommendations, we have added a new sub-section titled "Limitations" within the Discussion section of the manuscript. This allowed us to thoughtfully address any limitations or concerns raised by the reviewers, while also making it clear why the overall study design and findings remain sound. We believe this addition will provide valuable context for the readers. Furthermore, we have carefully considered and incorporated the comments highlighted at point 5 by each reviewer. This has resulted in several improvements throughout the manuscript. Notably, we have revised the tables and converted them to figures to enhance clarity and visual impact. Additionally, we made other minor edits to improve the overall quality and coherence of the work, including thoughtfully refining the language to ensure a more inclusive tone. We have also updated the title of the paper to "Overcoming the Tropical Andes Publication Divide: Insights from Local Researchers on Challenges and Solutions." This new title better reflects the qualitative nature of our study and its focus on providing in-depth insights from researchers in lower publication countries in the Tropical Andes. We appreciate your insightful feedback and believe these changes significantly enhance the manuscript. Please let us know if there are any other areas that require further adjustment. Specific responses to reviewers comments are below. 

 Reviewers' comments:

 Reviewer's Responses to Questions

Comments to the Author

 1. Is the manuscript technically sound, and do the data support the conclusions?

Reviewer #1: No

Reviewer #2: Yes

RESPONSE: We have addressed reviewer 1’s specific comments on this issue below.

2. Has the statistical analysis been performed appropriately and rigorously?

Reviewer #1: No

Reviewer #2: No

RESPONSE: We have not performed any statistical analyses as it is a quantitative study. We have addressed this in greater detail below.________________________________________

3. Have the authors made all data underlying the findings in their manuscript fully available?

Reviewer #1: No

Reviewer #2: No

RESPONSE: We have now included the anonymized dataset as supplementary material.

4. Is the manuscript presented in an intelligible fashion and written in standard English?

Reviewer #1: Yes

Reviewer #2: Yes

NO RESPONSE NEEDED.

5. Review Comments to the Author

Reviewer #1: Authors manuscript is well written and in standard English. However there are several flaws on the design of data adquisition. Authors based their results on people registered for a workshop which was advertised on social media. Authors do not compare their results for instance with the data from Peruvian researchers with CTI vitae from CONCYTEC (Consejo Nacional de Ciencia y Tecnología) which is an open source about researcher publications and other information.

RESPONSE: Thank you for your thoughtful comments and critique of our study. We appreciate the opportunity to address your concerns. While we acknowledge that this approach may introduce potential biases, it aligns with the specific objectives of our study. Our primary goal was not to obtain a strictly representative sample of the entire research community in the Tropical Andes region. Instead, we aimed to gather insights from researchers who were actively interested in and motivated to address the challenges associated with scientific publishing. The overwhelming response rate, with over 500 applicants expressing interest within a short period, demonstrates a substantial pool of researchers facing similar challenges and a widespread interest in the topic. We have now added this to the manuscript to make it clearer “The primary aim was to gather in-depth perspectives from those actively seeking support and guidance to overcome challenges in publishing their research.” Additionally, while comparing our results with databases such as CONCYTEC could provide additional perspectives, it was not within the scope or purpose of our study. Our research focused on gathering firsthand experiences and perspectives directly from researchers in the region through qualitative methods. Furthermore, the CONCYTEC database is specific to Peru, whereas our study aimed to capture insights from multiple countries within the Tropical Andes region, including Ecuador and Bolivia. However, we have now added a "Limitations of the Study" section in the Discussion which elaborates on this and other limitations.

 Their sample is very small and not random.

RESPONSE: We disagree with the characterization of our sample as very small. Our study included over 500 respondents, which provides a substantial dataset for qualitative analysis. However, we acknowledge that the sample is not random. It comprises researchers who expressed interest in a workshop on publishing, thus representing those particularly motivated to address publishing challenges. We have now made this clearer in the manuscript: “The primary aim was to gather in-depth perspectives from those actively seeking support and guidance to overcome challenges in publishing their research.” However, we have now added a "Limitations of the Study" section in the Discussion, as suggested. This section elaborates on the limitations picked up by the reviewers, including the non-random nature of our sample and the lack of statistical analyses, and explains why the study is still valuable despite these limitations.

There is not any statistical analyses so it is difficult to apply their results to a more general population, much less the tropical Andes

RESPONSE: While we agree that our study does not include statistical analyses, this is because it is a qualitative study by design. The focus is on gathering detailed insights and experiences from researchers rather than producing generalizable statistical data. Qualitative research is particularly suited to exploring complex issues in depth, capturing the nuances of participants' experiences, and providing insights into their perspectives. Our approach allowed us to identify specific barriers and needs that researchers in the region face, which might be overlooked in quantitative studies. 

It is also important to clarify that our study does not aim to generalize findings to the entire Andes region. As stated in the introduction, our focus was on Bolivia, Ecuador, and Peru, countries that publish the least in Latin America. We dedicated an entire paragraph mentioning Colombia, which has a significantly higher publication rate, as an outlier in the region. Therefore, our goal was not to generalize to the entire Tropical Andes but to focus on the specific context and challenges within these three countries in particular. 

Nevertheless, we have now changed the title of the paper to "Overcoming the Tropical Andes Publication Divide: Insights from Local Researchers on Challenges and Solutions." This new title better reflects the qualitative nature of our study and its focus on providing in-depth insights from researchers in lower publication countries in the Tropical Andes, rather than attempting to produce generalizable statistical data.

Additionally, we have made this more clear in the limitations section: “...the qualitative approach restricts the ability to statistically compare the findings with other studies or generalize the results to a broader population of the Tropical Andes. However, this approach aligns with the primary aim of gathering rich, contextual data from the targeted population of researchers in the Tropical Andes region who were actively seeking support for publishing their work. The intent of this study was to understand their specific perspectives and needs in overcoming barriers, rather than producing statistically representative findings. Despite these limitations, the study provides valuable insights that can inform future research and targeted initiatives to address the publishing challenges faced by this subset of the research community.”

 The criteria for their selection of registered researchers is not clear. Authors should have included the raw data as supplementary material. We do not know how many of the selected surveyed people work or are part of a university, national research institution, private research institution or are free researchers.

RESPONSE: We acknowledge that our initial explanation regarding the selection criteria for registered researchers may have lacked clarity. Our criteria aimed to assemble a diverse and experienced group of participants from various backgrounds, including academia, national research institutions, private research institutions, and independent researchers. Factors such as geographical origin, previous participation in relevant workshops, academic background, and expertise were carefully considered during the selection process. We have now provided a detailed explanation of our selection process in the Methodology section: “From the pool of over 500 registered applicants, we carefully selected 292 participants to take part in the workshop. The selection process focused on geographical origin (prioritizing the Tropical Andes region, particularly Peru, Bolivia, Ecuador, Colombia, and Venezuela), previous participation in related workshops and those on the ACCA mailing list (indicating relevant interest and engagement), age and academic background (excluding applicants 22 years old or younger, recent undergraduates within the last year and prioritizing those with postgraduate education), academic expertise (favoring ecology and conservation over communication and social sciences), English proficiency, and individuals with material ready for publishing. After thoroughly evaluating survey responses against these criteria, we were able to have a select group of participants with diverse expertise and experiences, enabling us to tailor the workshop to meet their needs and expectations.” We believe this detailed explanation enhances the transparency and understanding of our methodology for participant selection.

Additionally, while we had detailed information from the registration survey, maintaining participant anonymity was fundamental to fostering open and honest responses in our study. However, the inherent flexibility of our study design, allowing participants to freely enter and exit the workshop during the three days, presented challenges in accurately attendance and survey completion. Despite our efforts to ensure a representative sample from the registration demographics, it was not possible to know which of the registered individuals ultimately participated in the workshop, adding complexity to our analysis. This fluidity made it difficult to link workplace data collected during registration to specific participants. Given these constraints, we removed the demography of the selected participants section, which was only two sentences, from the results as we cannot confidently associate this information with the general demography of the individuals who ultimately participated in the workshops. We have made this clear in the limitations section: “While this study offers valuable insights into the challenges faced by individuals in the Tropical Andes region regarding scientific publishing, several limitations should be acknowledged. First, the registration survey was not originally designed for research purposes, but rather as a tool for selecting participants and designing the workshop. The open-ended nature of the questions required the data to be retrospectively categorized and coded, which may have introduced subjectivity. Additionally, while over half of the participants were selected for the workshop, the decision to maintain anonymity meant there is no record of attendance or way to link individual responses across the pre- and post-workshop surveys. Many participants could only attend for one day or leave before completing the surveys, further limiting the ability to capture complete data. This lack of identifying information also prevented exploration of potential differences in challenges based on factors like nationality, institution type, or career stage.” Lastly, in response to the reviewer’s suggestion, we have now added the anonymized survey data as supplementary material, removing any identifying information such as names and emails. This addition enhances the transparency and accessibility of our data.

Reviewer #2: Figure 2 does not specify which subplots correspond to A and B, and the legend of Figure 2B is small and challenging to read.

RESPONSE: We have now removed this figure as explained in the previous comment. 

 In Figure 3, we depict the primary findings derived from the survey responses. Given the nature of the data, employing a Pareto chart would be more appropriate for effectively highlighting the main reasons across all topics."

RESPONSE: The Pareto chart was not used because each participant could provide multiple responses. This would result in a cumulative percentage exceeding 100%, rendering the Pareto chart unsuitable for accurately representing the data.

Exploring nationality-based response variations could provide valuable insights. Utilizing a chi-square test for analysis, non-significant results may suggest similarities among the countries, while significant differences would indicate distinctions despite their shared continent.

RESPONSE: This is a great idea. However, while we initially included nationality questions in the pre-selection survey, we overlooked their inclus

---

## [Editor Report · Decision Letter 2]

13 Jun 2024

Overcoming the Tropical Andes publication divide: insights from local researchers on challenges and solutions

PONE-D-24-01605R2

Dear Dr. Valdez,

We’re pleased to inform you that your manuscript has been judged scientifically suitable for publication and will be formally accepted for publication once it meets all outstanding technical requirements.

Kind regards,

Umberto Baresi, Ph.D.

Academic Editor

PLOS ONE

---

## [Editor Report · Acceptance letter]

17 Jun 2024

PONE-D-24-01605R2 

PLOS ONE

Dear Dr. Valdez, 

I'm pleased to inform you that your manuscript has been deemed suitable for publication in PLOS ONE. Congratulations! Your manuscript is now being handed over to our production team.

Kind regards, 

on behalf of

Dr. Umberto Baresi 

Academic Editor

PLOS ONE